# Evaluation of Mitochondrial Dysfunction and Idebenone Responsiveness in Fibroblasts from Leber’s Hereditary Optic Neuropathy (LHON) Subjects

**DOI:** 10.3390/ijms241612580

**Published:** 2023-08-08

**Authors:** Mirko Baglivo, Alessia Nasca, Eleonora Lamantea, Stefano Vinci, Manuela Spagnolo, Silvia Marchet, Holger Prokisch, Alessia Catania, Costanza Lamperti, Daniele Ghezzi

**Affiliations:** 1Medical Genetics and Neurogenetics Unit, Fondazione IRCCS Istituto Neurologico Carlo Besta, 20126 Milan, Italy; 2Institute of Human Genetics, School of Medicine, Technical University of Munich, 81675 Munich, Germany; 3Institute of Neurogenomics, Helmholtz Zentrum München, 85764 Munich, Germany; 4Department of Pathophysiology and Transplantation (DEPT), University of Milan, 20122 Milan, Italy

**Keywords:** Leber’s hereditary optic neuropathy, LHON, mtDNA, idebenone, fibroblasts, biomarker, responsiveness

## Abstract

Leber’s hereditary optic neuropathy (LHON) is a disease that affects the optical nerve, causing visual loss. The diagnosis of LHON is mostly defined by the identification of three pathogenic variants in the mitochondrial DNA. Idebenone is widely used to treat LHON patients, but only some of them are responders to treatment. In our study, we assessed the maximal respiration rate (MRR) and other respiratory parameters in eight fibroblast lines from subjects carrying LHON pathogenic variants. We measured also the effects of idebenone treatment on cell growth and mtDNA amounts. Results showed that LHON fibroblasts had significantly reduced respiratory parameters in untreated conditions, but no significant gain in MRR after idebenone supplementation. No major toxicity toward mitochondrial function and no relevant compensatory effect in terms of mtDNA quantity were found for the treatment at the tested conditions. Our findings confirmed that fibroblasts from subjects harboring LHON pathogenic variants displayed impaired respiration, regardless of the disease penetrance and severity. Testing responsiveness to idebenone treatment in cultured cells did not fully recapitulate in vivo data. The in-depth evaluation of cellular respiration in fibroblasts is a good approach to evaluating novel mtDNA variants associated with LHON but needs further evaluation as a potential biomarker for disease prognosis and treatment responsiveness.

## 1. Introduction

Leber’s hereditary optic neuropathy (LHON) is a neurodegenerative disease of the optic nerve that leads to bilateral subacute visual loss in 1 in 31,000 people, mainly affecting young adults [1].

The most common genetic diagnosis of LHON is due to maternally inherited pathogenic mitochondrial DNA (mtDNA) variants [1]. Indeed, over 90% of LHON cases are the result of three missense point mutations, m.14484T>C, m.3460G>A and m.11778G>A; the latter is the most relevant, causing 60–80% of LHON cases worldwide [2]. All these variants affect genes encoding complex I subunits of the respiratory chain and alter evolutionarily conserved amino acids: m.3460G>A (p.Ala52Thr) in *MT-ND1,* m.11778G>A (p.Arg340His) in *MT-ND4* and m.14484T>C (p.Met64Val) in *MT-ND6* [1]. Several additional mtDNA variants have been described but are often present in single LHON cases/families; furthermore, for most of these variants, functional studies have not been performed and they are classified as variants of uncertain significance (VUS) [3]. No biomarkers have been established for the diagnosis of these cases, other than the typical LHON symptomatology investigation. Although LHON was historically considered a mtDNA disease, some nuclear genes (particularly *DNAJC30*) have been recently associated with phenotypes overlapping with LHON.

LHON is characterized by reduced penetrance, particularly for females; indeed, penetrance is approximately four times higher in males [4]. This phenomenon is not related to heteroplasmy or threshold effects, since the mtDNA variants causing LHON are usually homoplasmic. The mitochondrial (i.e., certain mtDNA haplogroups) and nuclear background [5], as well as exposure to environmental toxic factors, seems to increase the risk of visual loss, while estrogens have been proposed to play a protective role in females [6].

One of the first approved drugs for mitochondrial diseases was idebenone, used for the treatment of LHON patients [7]. Idebenone is a synthetic analogue of coenzyme Q that helps electron transfer along the respiratory chain. It can also behave as an antioxidant and free radical scavenger, although this role is debated [8]. This compound could be able to compensate for the complex I deficiency, or increased oxygen radical production, in LHON patients. The activation of idebenone seems to occur thanks to its reduction by the NAD(P)H oxidoreductase I (NQO1) enzyme; the reduced form can shuttle electrons directly to complex III (CIII), bypassing the defective CI [9,10,11]. Some isolated and retrospective studies support the notion that idebenone could be effective to partially restore vision and accelerate the recovery time in LHON patients [12]. However, some other studies question the clinical effect of idebenone treatment, struggling to distinguish a placebo from idebenone’s effects [13].

Other studies have sought a preclinical model that can predict the course of the disease. Fibroblast cell lines represent a well-studied preclinical model for defects that affect respiratory chain complexes and are available in most cases of mitochondrial disease patients [14]. The usefulness of cultured fibroblasts concerns the possibility of verifying and studying the pathogenetic mechanism of the disease and, possibly, evaluating the efficacy of individual therapeutic options. LHON-derived fibroblasts show variable and contradictory results in the evaluation of mitochondrial respiration [2,15] and also regarding the effects of in vitro treatment with idebenone [3,16]. 

In this study, we assessed mitochondrial respiration in fibroblasts harboring different mtDNA pathogenic variants associated with LHON with variable disease penetrance, and we used respiration values as an indicator of the pathological condition. Our aim was to try to identify some correlations between genetics, clinical phenotype, disease penetrance and respiratory activity in fibroblasts and to predict patients’ responsiveness to idebenone treatment through them.

## 2. Results

The evaluation of mitochondrial respiratory function was performed by measuring the oxygen consumption rate (OCR) in fibroblast cell lines cultured in standard glucose medium. We focused our evaluation on the maximal respiration rate (MRR) as an indicator of the overall mitochondrial respiratory chain activity of each cell line. The MRR is obtained using carbonyl cyanide-4-(trifluoromethoxy) phenylhydrazone (FCCP), which mimics the physiological energy demand, inducing the respiratory chain to work at maximum capacity to deal with this metabolic request. In naïve conditions, a statistically significant reduction in MRR (*p* < 0001) was observed in all eight LHON fibroblast cell lines compared to two wild-type fibroblast lines used as controls, with a mean MRR decrease of approximately 30% in LHON cells (Figure 1a). Notably, a similarly defective MRR was observed between LHON cells from affected and unaffected subjects (Figure 1b), irrespective of the genotype. The LHON cell lines also presented reduced values of basal OCR, before the injection of any compound or inhibitor, indicating defective respiratory activity and a reduced cell energetic demand under baseline conditions (Appendix A). The level of decrease in patients for both the MRR and basal OCR values was similar, leading to a minimal impact on spare or reserve capacity. Spare capacity, i.e., the difference between basal and maximal respiration, is an index of the cell response to an increased energy demand, and its increase may indicate a compensatory mechanism to allow more ATP production to overcome major mitochondrial dysfunction or stress. Furthermore, OCR after the injection of the ATP synthase inhibitor oligomycin was reduced compared to control cells; this parameter represents the portion of basal respiration used to drive mitochondrial ATP production in response to the energetic needs of the cell (Appendix A). 

According to previous publications [2,17], to assess the effect of idebenone on mitochondrial function in fibroblasts, we used 1 µM idebenone for 72 h. After this treatment, no significant variations were found in terms of MRR. No major differences could be observed among cell lines carrying different mtDNA variants (Figure 1c). By combining all mutant lines in a single average LHON model, the mean MRR value in mutant cells at basal conditions was 68 ± 19%, while an upward trend, corresponding to 74 ± 22%, was observed in the treated cells (Figure 1d). Although all comparisons between basal conditions and treatments were not significant, fibroblasts from LHON subjects responsive to idebenone treatment showed a greater increase after drug supplementation compared to cells from unresponsive patients (Figure 1d).

There are diverse polymorphisms in the NQO1 gene associated with decreased protein amounts or activity, such as rs1800566/p.Pro187Ser and rs1131341/p.Arg139Trp [18,19]. Given the importance of NQO1 for the function of idebenone, *NQO1* variants have been investigated as possible modulators of the efficacy of idebenone treatment [20]. We sequenced *NQO1* in our cohort and found that most of the LHON subjects were homozygous wild type; only two individuals were heterozygous for the c.559C>T/p.Pro187Ser polymorphism and one for the c.72G>A/p.Glu24Glu synonymous change (rs689453) [21] (Table 1). Hence, *NQO1* variants could not explain the absence of an idebenone response in our cells.

We then evaluated whether this drug’s supplementation could affect cellular health by examining cell growth. We performed a cell proliferation assay via the MTT test on two control cell lines and three representative LHON cells (one for each common LHON variant). The latter showed a partial reduction in growth rate compared to controls but we noted that all cell lines grew more slowly after idebenone supplementation (Figure 2a) compared to standard culture conditions.

To assess whether the effect of the drug on the cell growth impairment observed on the representative cell lines had an impact on mitochondrial function, we then assessed the MRR using different idebenone concentrations, from 0.5 to 10 µM. Since some studies have suggested adding a reducing agent, such as DTT, in the culture medium, to prevent the depolarizing effects of idebenone [24], we tested also this variable for the higher concentration of treatment. The mitochondrial respiration results showed that idebenone did not reduce the MRR at concentrations between 0.5 and 2 µM (1 µM was the concentration used in our previous experiments), with a partial deleterious effect only at 10 µM (Figure 2b). We did not observe any improvement in the MRR after DTT addition, but rather a damaging effect of this compound on the MRR when used in combination with idebenone or even alone (Figure 2b). Moreover, we evaluated the time extension of the idebenone (1 µM) treatment to one or two weeks. Even with an increased incubation time, no significantly increased MRR was present in either control cells or fibroblasts from LHON subjects. In fact, in some cell lines, a longer idebenone treatment was deleterious for the respiratory parameters (Figure 2c).

Finally, we measured the mtDNA amount as an index of mitochondrial biogenesis. An increased mtDNA copy number has been proposed as a protective factor in LHON mutation carriers, with the hypothesis that compensatory mitobiogenesis may explain incomplete penetrance. In basal conditions, mtDNA levels higher than those of controls were found in the cell lines of affected subjects #1 and #2 and the asymptomatic subject #6 (Appendix A, blue columns). For the two m.14484T>C cell lines, a form of compensation seemed to be present in line #6, from an unaffected mother, compared to cells #5, from her son, who was found to have lower mtDNA and presented with impaired vision. In this case, a protective role of mitochondrial biogenesis against the phenotype could be assumed. However, we did not have sufficient data to confirm this correlation. The same compensation effect could be hypothesized for sample #1, carrying the m.3460G>A variant. Concerning #2 and #3, harboring the m.11778G>A variant, there was no correlation between the mtDNA levels and disease penetrance, since the affected subject #2 had higher mtDNA levels than his asymptomatic mother #3. After idebenone treatment, only lines #3 and #7 showed an increase in mtDNA levels, of which only the last one was statistically significant (*p* < 0.05). These two patients had different genetic defects, m.11778G>A and m.14538A>G, respectively. Among the m.11778G>A cluster, only number #3 appeared to show mtDNA compensation after idebenone treatment. This may exclude a strict correlation between the effect of idebenone and specific genetic mtDNA variants. Moreover, no correspondence was evident between mtDNA compensation and any MRR increase following idebenone treatment.

Overall, the findings obtained in the fibroblasts from subjects #1–6, harboring the “classical” LHON variants (m.3460G>A, m.14484T>C and m.11778G>A), were comparable to those obtained in subjects #7–8, carrying rare or private pathogenic variants, including the only subject of the cohort with a heteroplasmic variant [14].

## 3. Discussion

LHON is mostly diagnosed by the identification of pathogenic variants that affect mtDNA [2]. Although most LHON cases harbor one of the three common pathogenic variants (m.14484T>C, m.3460G>A, m.11778G>A), further rare mtDNA variants have been described as possible causes of LHON. However, most of them are classified as VUS because they are present in single cases or small pedigrees, often with incomplete penetrance. 

From the perspective of discovering and confirming new mtDNA variants as causative for LHON, it would be important to identify biomarkers and functional studies that can support their pathogenic role. Fibroblast cell lines are well described as an easy preclinical model for respiratory chain complex deficiencies, since they recapitulate most of the biochemical defects and can be obtained by a minimally invasive procedure [14]. Furthermore, the different heteroplasmy levels between affected and non-affected tissues, which is a typical issue concerning the use of fibroblasts, is not relevant for LHON since most of the mtDNA variants associated with this condition are homoplasmic. 

However, a critical point for LHON regards the tissue specificity of this mitochondrial disorder with the selective neurodegeneration of a subset of eye cells (i.e., retinal ganglion cells). This aspect has always led to doubt regarding the possible use of skin fibroblasts for LHON studies. Previous papers have reported some contrasting results, with either normal or defective biochemical readouts (usually complex I activity or oxygen consumption) in cells from LHON patients [2,22,25]. Notably, these studies were conducted on single or very few cell lines.

In this regard, our study showed that all eight tested fibroblasts from individuals harboring different LHON pathogenic variants displayed impaired cellular respiration, regardless of disease penetrance. Therefore, our results indicate an impaired MRR (as well as basal respiration) as a potential biomarker to validate pathogenicity for novel mtDNA variants and to support the genetic diagnosis of LHON.

We investigated the MRR also as a prognostic biomarker of LHON by correlating its value with the genetic defects and clinical presentation of subjects harboring LHON-associated variants. No clear correlation between our MRR results and genetic assets was found, but our patient cohort was too small to perform a statistically significant correlation study.

Regarding therapy responsiveness, it is known that only a subset of LHON patients are responders to most of the used supplements, including idebenone [13,14,15]. In our study, we evaluated the therapy-predictive value of LHON fibroblasts by measuring the MRR after idebenone treatment. Some cell lines seemed to show an improved MRR after the treatment, although the gain was not statistically significant for all treated cell lines. Considering the average value of all eight mutant fibroblasts compared to the controls, we observed a slight increase in the idebenone-treated cells, which did not reach statistical significance. Interestingly, the upward trend was greater in fibroblasts from LHON subjects responsive to idebenone. There are some critical points to note on this issue regarding the eight LHON individuals enrolled because (i) some subjects (the asymptomatic ones) did not receive any therapy and (ii) the affected subjects (responsive or not to idebenone) did not receive the same treatment in terms of dosage and duration. This complicated the clinical assessment of the responsiveness to idebenone, both because of differences in their therapies and because patients manifested initial symptoms of variable severity.

The conditions used for fibroblast treatment with idebenone in this study (1 µM for 72 h) had an impact on cell growth but idebenone did not reduce the MRR at up to 2 µM; hence, we can exclude any toxic effect on respiratory activity for the used concentrations. A role for two common *NQO1* polymorphisms in modulating the efficacy of idebenone treatment was excluded. Moreover, idebenone treatment did not cause a consistent change in mtDNA amount; therefore, we did not observe a correlation between idebenone’s effects and any compensatory mitobiogenesis mechanisms after the treatment.

## 4. Materials and Methods

We analyzed primary fibroblast cell lines isolated from 8 subjects harboring mtDNA variants associated with LHON. These subjects had different genetic variants and presented with variable symptomatology and disease penetrance. Some of them were also related by parenthood. Each patient received idebenone treatment with variable posology.

Table 1 reports a summary of the demographic, genetic and clinical data of the subjects enrolled in this study. All selected subjects harbored mtDNA variants associated with LHON, either the classical mutations (subjects #1–6) or rare likely pathogenic variants (#7 and #8), being all these variants in the homoplasmic state, except for subject #8, who presented a high heteroplasmy level (91% in fibroblasts). The affected subjects presented with a classical LHON phenotype: bilateral painless vision loss together with typical visual field defects and thinning of the retinal nerve fiber layer detected by optical coherence tomography. This resulted in reduced visual acuity of variable but similar degree. Two asymptomatic mothers (#3 and #6, homoplasmic for the same pathogenic variants as their affected offspring) were also included.

The *NQO1* genetic test was carried out by Sanger sequencing for all 6 exons of the gene, including the previously studied SNPs rs1800566 (c.559C>T/p.Pro187Ser) [18] and rs1131341 (c. 465C>T/p.Arg139Trp) [19]. Skin biopsies were taken from all subjects following standard clinical procedures. Fibroblast cell lines were cultured in Dulbecco’s Modified Eagle’s Medium (DMEM; ECB7501L EuroClone) containing 4.5 g/L of glucose and supplemented with 10% fetal calf serum, 1 mM sodium pyruvate, 200 U/mL penicillin G, 200 mg/mL streptomycin and 4 mM glutamine. Cells were grown at 37 °C under 5% CO_2_ conditions. Idebenone was previously prepared by suspending a powder (I5659; Sigma-Aldrich) in methanol and then diluting it in the culture medium to obtain 1 µM as a final concentration, for 72 h of treatment [2]. In all treated cells, fresh idebenone-supplemented medium was used every 24 h. A titration experiment was also conducted by using different concentrations of idebenone (from 0.5 to 10 µM) to test toxicity and/or activity on two LHON cell lines representative of the m.11778G>A and the m.14484T>C mutations, and in combination with dithiothreitol (DTT) as a reducing agent to prevent possible idebenone toxicity [22]. DTT was tested alone at 1 mM. Moreover, different incubation times (1 and 2 weeks) with 1 µM idebenone were tested.

The evaluation of mitochondrial respiration was carried out using the Seahorse XFe96 Analyzer (Agilent, Santa Clara, CA, USA). The oxygen consumption rate (OCR) was measured in basal conditions (basal respiratory activity) and after sequentially adding oligomycin (10 µM), carbonyl cyanide-4-(trifluoromethoxy) phenylhydrazone (FCCP, 5 µM) and finally rotenone (20 µM) plus antimycin A (25 µM); the OCR was used for parameter calculations [26]. Fibroblasts were seeded in Seahorse 96-well plates at a concentration of 2 × 10^4^ cells/100 μL per well, and eight replicates for each cell line were performed in every single experiment. At the end of the Seahorse experiment, we used the CyQUANT™ Direct Cell Proliferation Assay (C35011, Invitrogene, Thermo Fisher Scientific, Waltham, MA, USA) to normalize the obtained OCR data to the total number of cells in each well. At least four independent Seahorse experiments were performed for each cell line. We evaluated the maximal respiration rate (MRR), i.e., the maximum rate of respiration that the cell could achieve, obtained by subtracting the maximal OCR measurement after FCCP injection and the minimum OCR measurement after oligomycin injection. Final reported MRR values were calculated as the average percentages of the maximal respiration rates from all the Seahorse experiments. We set the value of one hundred percent for the average of the percentage of two wild-type fibroblast cell lines that we used as controls in each experiment. A comparison between the MRRs of these two controls and other control cell lines is reported in the Appendix A. Student’s t-test was performed as a statistical analysis to evaluate significant differences between results arising from two or more conditions.

The cell proliferation assay was performed using an MTT kit (CT02; Millipore, Burlington, MA, USA). We performed the MTT test on two control cell lines and three representative LHON fibroblast cell lines at four different time points every 24 h, both for idebenone treatment and untreated conditions.

To quantify mtDNA, we collected cell pellets for idebenone-treated and untreated conditions, after 48 h of culture, and extracted the total DNA using the QIAamp DNA Micro Kit from Qiagen. Real-time quantitative PCR (qPCR) was performed by using the SsoAdvanced Universal Probes Kit (BioRad, Hercules, CA, USA), with primers and probes for the 12S subunit of mitochondrial rRNA and the TaqMan RNAse P Control Reagent Kit (Applied Biosystems, Thermo Fisher Scientific, Waltham, MA, USA) as a nuclear reference [27]. We performed at least three independent qPCR analyses for each cell line and for each condition. We normalized these results as the relative amount of mtDNA compared to the two control cell lines.

## 5. Conclusions

In summary, our study aimed to promote research on the cellular respiration of fibroblasts as a biomarker for the diagnosis of LHON. The early diagnosis of LHON is essential for the treatment of this debilitating disease. Unlike the variable and contradictory results reported in the scientific literature, our data support the notion that fibroblasts from LHON patients consistently display impaired mitochondrial respiration, regardless of disease penetrance or severity. Therefore, the MRR in fibroblasts could be a good biomarker for LHON diagnosis and also for the evaluation of novel mtDNA variants associated with LHON. Based on our cohort, it is not possible to discern clear genetic or clinical correlations. Indeed, different genetic variants and differences in the severity of the phenotype (including asymptomatic status) led to similar MRR impairments.

Using the selected cell lines and experimental settings, it was not possible to establish fibroblasts as a completely reliable model in assessing/predicting idebenone responsiveness. However, we observed an upward trend in idebenone-treated conditions, more evident in cells from idebenone-responsive patients. This difference was not significant, possibly due to the small LHON groups or the marginal benefits on the tested readouts, not measurable by the instrument. Several additional alterations (increased ROS production, altered membrane potential, impaired cell growth, reduced ATP production) have been reported in different cell models of LHON and could be further investigated. Nevertheless, having an in vitro phenotype to follow is a good starting point to evaluate the responsiveness to treatments.

An in-depth evaluation of cellular respiration in fibroblasts could provide valuable insights into the diagnosis and treatment of this disease, acting as a potential biomarker for LHON and its genetic predisposition to idebenone responsiveness.

## Figures and Tables

**Figure 1 ijms-24-12580-f001:**
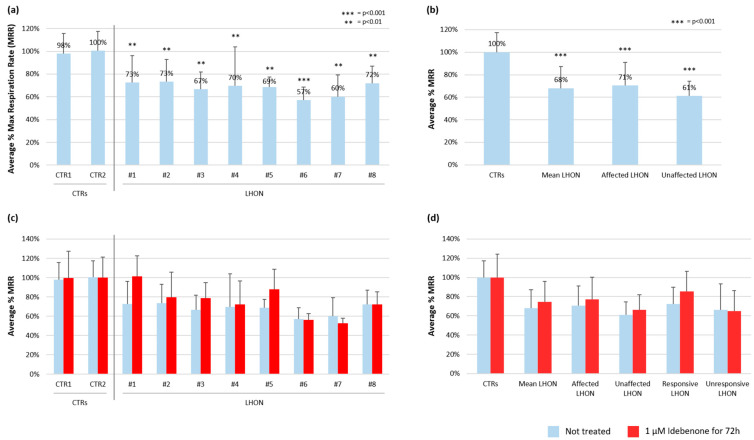
(**a**) Percentage of maximal respiratory rate (MRR) in 8 LHON fibroblast cell lines compared to two controls (CTR), in standard culture media. The average MRR of the 2 CTRs was set as 100% (*** = *p*-value < 0.001, ** = *p*-value < 0.01). (**b**) Percentage of MRR mean value in control and LHON cells; the affected LHON group consisted of all LHON analyzed, excluding asymptomatic subjects #3 and #6, who constituted the unaffected group. (**c**) Percentage of MRR before (blue columns) and after (red columns) idebenone treatment (1 µM for 72 h). (**d**) Percentage of mean MRR in basal conditions and after idebenone treatment in control and LHON cells, clustered according to their disease penetrance and responsiveness to treatment. The responsive LHON group included subjects #1, #2, #5 and #8, who had improvements after idebenone therapy; subjects #4 and #7, who followed the therapy without an improvement, constituted the unresponsive group.

**Figure 2 ijms-24-12580-f002:**
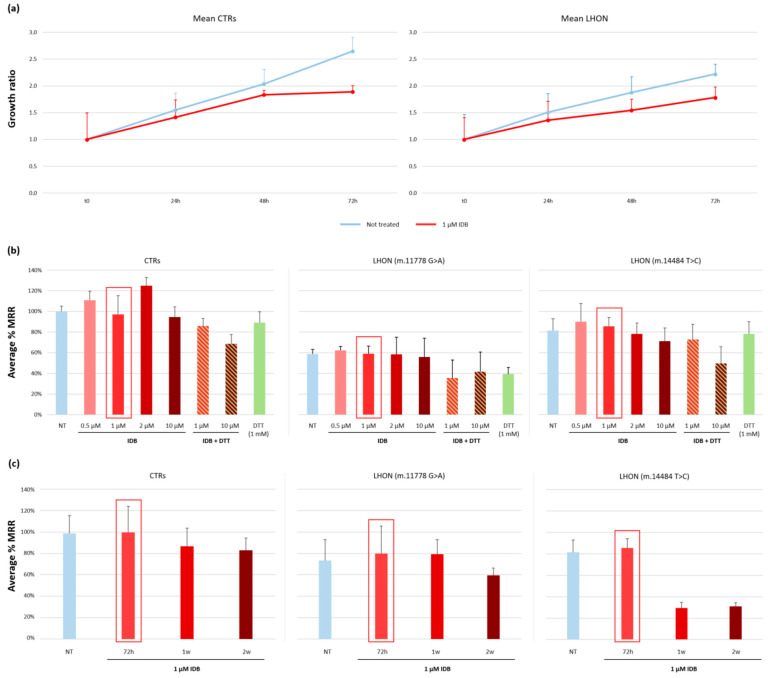
(**a**) Growth ratios of control and LHON cell lines, untreated (blue lines) and treated with 1 µM idebenone (red lines) for 24–48–72 h, measured by the MTT test. CTRs were two control cell lines, while LHON cells were three lines, one for each common LHON variant. (**b**) MRR measured in two control (CTRs) and two LHON cell lines (harboring the m.11778G>A and the m.14484T>C variants, respectively) in basal conditions (blue columns, set as 100% for CTRs) and for idebenone titration from 0.5 to 10 µM for 72 h of treatment (red columns). Idebenone concentrations 1 µM and 10 µM were evaluated without (red columns) and with (green/red striped columns) addition of DTT as a reducing agent. DTT 1 mM was tested also alone (green columns). (**c**) MRR measured in two control (CTRs) and two LHON cell lines (harboring the m.11778G>A and the m.14484T>C variants, respectively) in basal conditions (blue columns, set as 100% for CTRs) and for different incubation times (from 3 days to 2 weeks; red columns) with 1 µM idebenone. Experiments were performed in duplicate.

**Table 1 ijms-24-12580-t001:** Genetic, demographic and clinical data of the investigated subjects.

#	mtDNA Variant (Gene)	Homoplasmy/Heteroplasmy	*NQO1* Sequencing	Age	Gender	Parenthood	Age at Diagnosis	Age at Biopsy	Visual Acuity (Right)	Visual Acuity (Left)	Idebenone Therapy	Vision Recovery
1	m.3460G>A (*MT-ND1*)	homoplasmic	c.559C>T heteroz	45	F	-	28	37	CF 1m *	CF 1m *	Yes	Yes
2	m.11778G>A (*MT-ND4*)	homoplasmic	c.559C>T heteroz	36	M	Son of #3	25	28	1	1	Yes	Yes
3	m.11778G>A (*MT-ND4*)	homoplasmic	wt	59	F	Mother of #2	/	50	Asymptomatic		No	/
4	m.11778G>A (*MT-ND4*)	homoplasmic	c.72G>A heteroz	22	M	-	16	20	CF 1m *	CF 1m *	Yes	No
5	m.14484T>C (*MT-ND6*)	homoplasmic	wt	30	M	Son of #6	29	19	1.3	0.5	Yes	Yes
6	m.14484T>C (*MT-ND6*)	homoplasmic	wt	62	F	Mother of #5	/	52	Asymptomatic		No	/
7	m.14538A>G ^ (*MT-ND6*)	homoplasmic	wt	48	M	-	38	39	1.3	1	Yes	No
8	m.14465G>A ^ (*MT-ND6*)	heteroplasmic 91%	wt	34	M	-	30	30	1	1	Yes	Yes

* CF 1m: counting fingers at 1 m. It is a semiquantitative scale for visual acuity (VA), used in patients with very low vision. ^: variants previously reported as likely pathogenic (m.14538A>G: p.Phe46Leu; m.14465G>A: p.Thr70Ile) [22,23]. Heteroz: heterozygous; wt: wild type; M: male; F: female.

## Data Availability

Data are available at the repository Zenodo: https://zenodo.org/record/7937991 (accessed on 1 August 2023). The other data that support the findings of this study are available from the corresponding author upon reasonable request.

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
