# Peer review of "Evaluation of Mitochondrial Dysfunction and Idebenone Responsiveness in Fibroblasts from Leber’s Hereditary Optic Neuropathy (LHON) Subjects"

_ijms, 2023, doi:10.3390/ijms241612580_

Round 1
Reviewer 1 Report
This paper describes poor quality results obtained with fibroblasts from various LHON patients.
The overall experimental design is flawed (why Methods are in section 4 and not upfront after the Introduction?), besides the poor quality of measurements. TWO controls are certainly not enough to evaluate the variability of Seahorse OCR measurements. The Authors appear to have little experimental background in mitochondrial biochemistry and this has seriously affected not only the experimental design, but also the quality of results presentation.
Moreover, there is no mention in the paper of the crucial role that cytosolic NAD(P)H oxidoreductase I (NQO1) has on the cellular activities and consequently clinical effects of idebenone.
Haefeli, Roman H., et al. "NQO1-dependent redox cycling of idebenone: effects on cellular redox potential and energy levels." PloS one 6.3 (2011): e17963.
The values of OCR in various LHON patients vary depending upon the specific mtDNA mutation they harbor; hence, there is no sense in averaging such data from different LHON patients together! Nevertheless, the apparent lower OCR values of patients harboring the common ND4 mutation most likely derive from artifacts or poor measurements - the previous literature failed to detect a significant decrease in electron transport activity of complex I.
Bottom line: this work cannot be published in a serious journal.
The English style should be improved. One detail gives away unfamiliarity with scientific English: 0,5 instead of 0.5
Author Response
>We thank the Reviewers for their comments and suggestions, which have prompted us to improve the quality and clarity of the paper.
Here, in blue there are our response.
Reviewer 1
This paper describes poor quality results obtained with fibroblasts from various LHON patients.
The overall experimental design is flawed (why Methods are in section 4 and not upfront after the Introduction?), besides the poor quality of measurements. TWO controls are certainly not enough to evaluate the variability of Seahorse OCR measurements. The Authors appear to have little experimental background in mitochondrial biochemistry and this has seriously affected not only the experimental design, but also the quality of results presentation.
>Our communication wants to present some data on cellular respiration of fibroblasts as a biomarker for the diagnosis of LHON. Despite variable and contradictory result reported in literature, our cohort consistently display impaired mitochondrial respiration, regardless of disease penetrance or severity. This report provides valuable insights into the diagnosis and treatment of this disease which certainly requires more in-depth study with larger numbers.
Methods were inserted in section 4 according to the IJMS Microsoft Word template file used to prepare the manuscript (https://www.mdpi.com/journal/ijms/instructions#submission). However, we noticed now that the MDPI instructions report a different sequence for the sections: Abstract, Keywords, Introduction, Materials and Methods, Results, Discussion, and Conclusions (optional). As suggested, we moved the M&M section after the Introduction.
Contrary to what the reviewer claims, our laboratory has many years of experience with mitochondrial biochemistry and the use of Seahorse instrument (see the paper Invernizzi et al. 2012 which was one of the first methodological papers about the use of the Seahorse instrument).
Regarding the control cell lines, we are aware of the variability of Seahorse OCR measurements.
We would like to reiterate that the results are from the mean of 4 or more independent experiments involving the TWO control lines which have been analyzed contextually with LHON cells. Because of the known variability of the instrument, we always prefer to analyze our patients compared to controls run on the same plate. From a practical perspective, it is not possible to include more than two controls in each plate.
To support the reliability of the selected lines (CTR1 and CTR2), we compared them to other 5 fibroblast lines (both healthy controls and unaffected carriers of single heterozygous variant responsible for recessive disorders). In contrast to LHON cells showing defective MRR, all other “control” cells showed values close to those observed in CTR1 and CTR2. These data are attached as supplementary figure S1.
Moreover, there is no mention in the paper of the crucial role that cytosolic NAD(P)H oxidoreductase I (NQO1) has on the cellular activities and consequently clinical effects of idebenone. Haefeli, Roman H., et al. "NQO1-dependent redox cycling of idebenone: effects on cellular redox potential and energy levels." PloS one 6.3 (2011): e17963.
>According to the reviewer’s suggestions, we added some sentences about the role of NQO1.
Given the importance of NQO1 for the function of idebenone, multiple polymorphisms in the NQO1 gene, present in the general population, have been investigated. Probably the most intesively studied is c.559C>T /p.Ser187 which is associated with 50% protein reduction in heterozygous carriers and with almost complete absence of NQO1 in homozygous carriers [Guha et al. 2008]. The same phenomenon was reported also for the SNP p. Arg139Trp [Singh et al. 2009]. Notably, at a recent congress on mitochondial medicine (Euromit2023, June 2023 in Bologna, Italy), a poster by Del Dotto et al. “Genetic variants impact on NQO1 expression and activity driving efficacy of idebenone treatment in Leber’s hereditary optic neuropathy cell models” evaluated the effects of these two SNPs in LHON cellular models.
Hence, we sequenced the NQO1 gene in our patients. However, most of our LHON subjects were homozygous wild-type for these SNPs, and two heterozygous for only one of them: hence, NQO1 variants (in particular, the two previously investigated variants ) cannot explain the absence of idebenone response in our cells. We added the info about NOQ1 polymorphisms in the text and in Table 1.
The values of OCR in various LHON patients vary depending upon the specific mtDNA mutation they harbor; hence, there is no sense in averaging such data from different LHON patients together! Nevertheless, the apparent lower OCR values of patients harboring the common ND4 mutation most likely derive from artifacts or poor measurements - the previous literature failed to detect a significant decrease in electron transport activity of complex I.
>We reported in the graph the single values from each LHON patient. However, in order to try to highlight a correlation between the MRR value and any genetic or phenotypic parameter, we clustered the patients into homogeneous group (affected, unaffected, responsive to treatment...). This is also necessary to increase the statistic power and to avoid the comparison 1 line vs 1 line.
Even the grouping by genotype (e.g., the common ND4 pathogenic variant) could be questionable, since mtDNA haplogroup could be different (in addition to several nDNA variant, including NQO1).
As stated in the introduction, data about the biochemical defects (complex I activity or oxygen consumption) in LHON patients are contradictory. As also present in Genereviews (Yu-Wai-Man et al. 2021), in vitro assays showed variable results: for instance, regarding the m.11778G>A, complex I activity is reported 0%-50% less than controls (i.e. either normal or defective), and respiratory rates decreased by 30-50% relative to controls. In our experience on LHON fibroblasts we observed the same findings: complex I is occasionally defective (without a strict link with a specific phenotype although m.3460G>A seems the worst). Conversely, as described in this manuscript, reduced maximal respiration is a common feature in all tested LHON lines.
Bottom line: this work cannot be published in a serious journal.
>In this regard, we let the editor decide (also considering the comments by the other reviewers who gave a different evaluation of our work compared to reviewer #1)
This work is a communication about what we found evaluating cellular respiration in a group of LHON fibroblasts. We tried to present both “positive” findings (a defective in vitro phenotype that may represent e a good starting point to evaluate responsiveness to treatments) and “negative” results (absence of idebenone effect on respiratory parameters, at least with tested experimental conditions).
Reviewer 2 Report
This manuscript by Balivo et al. is a straight-forward investigation that uses LHON mutation fibroblasts to examine oxygen consumption rate and idebenone supplementation. The results look good and the interpretation of the results are consistent with their findings. I only have a few minor points:
How long were the cells allowed to grow after seeding them onto the Seahorse 96-well plates? Did the authors normalize the OCR to the total number of cells? As they show that there is a difference in growth rates in Figure 2A, it would be important to normalize the OCR rates to growth rate as the reduced growth rate could explain some of the difference in MMR.
It would help the reader if the authors quickly described the effects of the various SNPs on the protein (e.g. what is the amino acid change and known repercussions of the change).
Author Response
>We thank the Reviewers for their comments and suggestions, which have prompted us to improve the quality and clarity of the paper.
Here, in blue our response.
Reviewer 2
This manuscript by Balivo et al. is a straight-forward investigation that uses LHON mutation fibroblasts to examine oxygen consumption rate and idebenone supplementation. The results look good and the interpretation of the results are consistent with their findings.
I only have a few minor points:
How long were the cells allowed to grow after seeding them onto the Seahorse 96-well plates? Did the authors normalize the OCR to the total number of cells? As they show that there is a difference in growth rates in Figure 2A, it would be important to normalize the OCR rates to growth rate as the reduced growth rate could explain some of the difference in MMR.
It would help the reader if the authors quickly described the effects of the various SNPs on the protein (e.g. what is the amino acid change and known repercussions of the change).
>Here the answers to Reviewer2’s minor points:
-our Seahorse protocol requires that the cells are counted and seeded in the 96-well plate the day before the experiment in the Seahorse instrument. Therefore, the cells grow in the SH 96-well plate for approximately 16/18 hours before Seahorse measurements. We seeded 20'000 cells/wells for each line. To take into account possible differences in growth rates, or counting errors, at the end of the Seahorse experiment we used CyQUANT™ Direct Cell Proliferation Assay (C35011, Invitrogene) as normalization method to normalize the obtained OCR data to the total number of cells in each well. This normalized value is used for further counting (e.g., MRR).
-we added a sentence about the predicted effect of the various missense variants in the introduction.
Reviewer 3 Report
In this manuscript, Baglivo et al assessed the maximal respiration rate (MRR) and other respiratory parameters in 8 fibroblast lines from subjects carrying LHON pathogenic variants. They also measured the effects of idebenone treatment on cell growth and mtDNA amount.
They worked with fibroblasts cell lines #1, #2, #5 and #8 from patients who had improvement after idebenone therapy, and fibroblasts cell cultures #4 and #7 from subjects who did not respond to idebenone treatment.
Results showed that LHON fibroblasts have significant reduced respiratory parameters in untreated conditions, but no significant gain in MRR after idebenone supplementation. They confirmed that fibroblasts from subjects harboring LHON pathogenic variants display impaired respiration, regardless of the disease penetrance and severity. Testing responsiveness to idebenone treatment in cultured cells did not fully recapitulate in vivo data. The authors conclude that patient-derived fibroblast cell cultures are a good approach for evaluating novel mtDNA variants associated with LHON but needs further evaluations as potential biomarker for disease prognosis and treatment responsiveness.
Suggestions for improving the manuscript:
1. Time of incubation with idebenone can be an important parameter that should be also evaluated.
2. It would be of interest to evaluate the potential positive effect of idebenone on other mitochondrial alterations (such as ROS production…) or protective mechanisms (mtUPR).
Author Response
>We thank the Reviewers for their comments and suggestions, which have prompted us to improve the quality and clarity of the paper.
Here in blue our response.
Reviewer 3
In this manuscript, Baglivo et al assessed the maximal respiration rate (MRR) and other respiratory parameters in 8 fibroblast lines from subjects carrying LHON pathogenic variants. They also measured the effects of idebenone treatment on cell growth and mtDNA amount.
They worked with fibroblasts cell lines #1, #2, #5 and #8 from patients who had improvement after idebenone therapy, and fibroblasts cell cultures #4 and #7 from subjects who did not respond to idebenone treatment.
Results showed that LHON fibroblasts have significant reduced respiratory parameters in untreated conditions, but no significant gain in MRR after idebenone supplementation. They confirmed that fibroblasts from subjects harboring LHON pathogenic variants display impaired respiration, regardless of the disease penetrance and severity. Testing responsiveness to idebenone treatment in cultured cells did not fully recapitulate in vivo data. The authors conclude that patient-derived fibroblast cell cultures are a good approach for evaluating novel mtDNA variants associated with LHON but needs further evaluations as potential biomarker for disease prognosis and treatment responsiveness.
Suggestions for improving the manuscript:
- Time of incubation with idebenone can be an important parameter that should be also evaluated.
>The possible combinations of different culture conditions (idebenone concentration, time, medium...) are myriad. However, based on reviewer’s comment, we tested different times of incubation for the fixed concentration of 1 uM idebenone, increasing it to 1 and 2 weeks. We selected two LHON cell lines for this evaluation, as done for testing different idebenone concentrations or DTT (figure 2). We observed that, even with increased incubation time, not significantly increased MRR was present in either control cells or fibroblasts from LHON patients. Really, in some cell lines long idebenone treatment resulted deleterious.
- It would be of interest to evaluate the potential positive effect of idebenone on other mitochondrial alterations (such as ROS production…) or protective mechanisms (mtUPR).
>We thank the reviewer for the good suggestion that will be further evaluated in a future study. Indeed, in addition to respiratory defects, several other alterations have been reported in different cell models of LHON. Increased ROS production is one of them, but also altered membrane potential, impaired cell growth, reduced ATP production, plus a series of additional dysfunctional pathways more recently reported: mitophagy, mitobiogenesis, inflammatory cytokines production...
Since the genetic defect in LHON patients affect the respiratory chain (complex I), we focused on an assay for the functionality of this system that could be used for a rapid screening of the functional effects of mtDNA LHON variants in patients’ fibroblasts. Then, to better interpret the obtained findings, we coupled this main assay with some tests on cell growth and mtDNA amount.
We agree that further parameters could be assessed, and we plan to extend the characterization of these LHON cells in future studies.
Round 2
Reviewer 1 Report
Although I have appreciated the additional data now added to the revised manuscript, especially on the genetics of NQO1, the results remain of questionable quality and value. Data are inconclusive and in part contradict solid biochemical and cellular data reported previously on LHON mutations. Maybe the article would fit in a clinically oriented journal, but not on a molecular journal.
The English style is poor
Author Response
We thank the Reviewer for his/her comments, which have prompted us to improve the quality and clarity of the paper.
On the final judgment, we let the editor decide (also considering the comments of the other reviewers who gave a different evaluation).
Reviewer 3 Report
The authors have addressed my concerns
Author Response
We thank the Reviewer for his/her comments.